# A Parental Behavior Scale in Pediatric Dentistry: The Development of an Observational Scale

**DOI:** 10.3390/children10020249

**Published:** 2023-01-30

**Authors:** Tania Vanhée, Farah Dadoun, Astrid Vanden Abbeele, Peter Bottenberg, Wolfgang Jacquet, Isabelle Loeb

**Affiliations:** 1Department of Dentistry, Faculty of Medicine, Université Libre de Bruxelles, 1070 Brussels, Belgium; 2Department of Surgical Clinical Sciences CHIR-ORHE, Faculty of Medicine and Pharmacy, Vrije Universiteit Brussel, 1090 Brussels, Belgium; 3Department of Educational Sciences EDWE-LOCI, Faculty of Psychology and Educational Sciences, Vrije Universiteit Brussel, 1050 Brussels, Belgium; 4Department of Stomatology and Maxillo-Facial Surgery, CHU Saint-Pierre, Université Libre de Bruxelles, 1000 Brussels, Belgium

**Keywords:** parent–child relations, parental influence, dental anxiety, dental fear, video recording, dental care for children

## Abstract

Children’s treatment means a triangular relationship between the child, practitioner, and parent, with specific interactions influencing the procedure. The objective was to create and validate a hetero-rating scale of parental behavior and verify the correlation between child and parental behavior during pediatric dentistry sessions. Treatment sessions were recorded and evaluated, including 60 children representing three age groups. Two raters interpreted the resulting video clips using the modified Venham scale for children and the new hetero-rating scale for parents. They analyzed the videos twice and attributed scores at different time points of the appointment. The correlation between parental behavior upon entrance and the children’s behavioral at the dental office in the treatment stage was significantly positive in both raters (Kendall Tau: 0.20–0.30). Furthermore, a panel of 20 dental practitioners scored a randomized selection of five recordings per age group. The level of agreement between the two experts was higher than that between the 20 clinicians. Venham types of scale involving multiple aspects can be used in research, but their application in dental practice requires further development. The link between parental anxiety and child anxiety is confirmed, but further research is required to incorporate specific aspects of treatment and parental behavior.

## 1. Introduction

The dental office generates anxiety and stress for many patients [1]. Anxiety is one of the primary human emotional expressions that can be encountered in a variety of situations, including dental treatment [2]. Dental anxiety stimuli change with age [3]. While knowing and understanding what lies ahead helps to reduce stress, the younger the child is, the more communication bias can take root [4]. Indeed, exaggerated dental anxiety is one of the most frequently diagnosed expressions in children [5]. Its estimated prevalence varies from 4% to 43% depending on the population [6]. This can often lead to disproportionate reactions, representing one of the greatest challenges faced by practitioners [7]. These disruptions can decrease the quality of oral health care services and cause future avoidance of dental visits [8,9]. Additionally, the acceptance of care prevents extreme situations of oral health neglect [6].

Since anxiety can act as an obstacle to the successful completion of care, it is fundamental that practitioners learn to recognize and manage the child’s anxiety [6]. First, the anxious behavior of the patient needs to be recognized in order to enable the practitioner to adapt to it and then to establish a sphere of trust [8].

The etiology of fear and anxiety in the chair is mainly rooted in apprehensions about dental treatment [10]. Very often, these apprehensions are created by the parents. Children witness their parents’ emotional reactions to dental care. In particular, parents showing a high level of anxiety themselves exert a negative influence on their children [11,12].

Parents have an even higher influence on children without previous dental experience because these children have not yet lived through their own experience [13]. Some parents even tend to use a dentist appointment as a punishment or threat, while others attribute a disproportionately heroic characteristic to it [8]. A previous painful dental experience coupled with increased parental dental anxiety may explain fear of the dentist in some children [6]. Indeed, negative dental experiences and a family history of anxiety are predictive of a child’s development of dental anxiety [9]. As a result, these fears are transferred from one generation to the next, creating a cycle of fear and distress from early childhood. When these feelings prevail in adulthood, they will again be reproduced and transferred to future generations [8,14].

While the etiology of the anxiety lies in interactions and anticipations before the actual session, the successful management of anxiety during the session is a result of the interactions between the parent, if present, and the patient and the attitude of the dental practitioner. Note that many practitioners indicate that they prefer to work in the absence of parents to avoid the projection of their anxiety, as well as their fears, onto their children [10]. However, children’s treatment means a triangular relationship between the child, practitioner, and parent, with specific interactions influencing the procedure [15].

In order to assess these influences, we chose to analyze the behavior of children and parents during dental procedures using behavioral rating scales.

Among the methods for evaluating the behavior of children during a dental session, the modified Venham scale (VS) is one of the most frequently used [16]. However, this scale has no equivalent for the parents accompanying the children. In what follows, we discuss a hetero-evaluation scale for parental behavior, which we created and validated as a basis for the study of parental behavior and its influence on child anxiety during a session at the dental office based on the scoring of video recordings. Furthermore, the partnership between the parent and practitioner in the management of child behavior during dental care is explored.

## 2. Materials and Methods

### 2.1. Study Design

#### 2.1.1. Evaluation Scale Design

The parental hetero-assessment scale was created through expert discussion based on four scales found in the literature: the VS [16] (see Table 1), ECPA (Evaluation Comportementale de la douleur chez la Personne Âgée—Behavioral Pain Scale in the Elderly) [17], PCS (Parent Cooperation Scale) [18], and CAMPIS (Child–Adult Medical Procedure Interaction Scale) [19,20]. The statements composing the instruments were recombined into five categories forming an ordinal scale in ascending order, according to the level of anxiety observed in the parents through given indicators (see Table 2). In the development phase, we sought to ensure the method’s validity through assessment and feedback provided by two senior academic experts among the authors (A.V.A. and I.L.). The choice to adopt a five-level scale was based on the nuance between the Uneasy and Tense categories, which was not deemed noticeable in the case of the parent’s behavior. Indeed, the main difference in the Venham scale is the slightly higher intensity of the child’s stress, observed on the levels of verbal and non-verbal behavior. It was difficult to apply this nuance to the assessment of parental behavior, as the descriptions were too close to differentiate them as two distinct relevant categories.

#### 2.1.2. Data Collection

In this study, a sample consisting of 60 patients presenting for pediatric dentistry consultation at the CHU Saint-Pierre César de Paepe (Brussels center) site for the first time were recruited. Sample size was estimated based on a study in Belgian children [21] thereby exceeding the *n* = 40 from a previous study on a related topic [8]. Three groups of 20 children aged 4 years and under, from 5 to 6 years old, and over 7 years old, respectively, were formed. These children were treated by two practitioners (T.V. and F.D.) between 11 January and 3 March 2021. The performed procedures were either care without local anesthesia, oral examinations with or without prophylactic cleaning, or treatments with local anesthesia, such as restorative care or extractions. The inclusion criteria were age, the ability to communicate in the language of the study, namely French, and acceptance to participate in the study. The exclusion criteria were ages falling outside the age groups in the study, refusal to participate, refusal to give access to image rights, and the presence of a mental or motor deficiency in the patient. Information about the study was provided orally and in writing via an information letter addressed to the parents and child. The informed consent document was presented and signed by the parent/caregiver of the child. Before entering the treatment room, each patient’s dental anxiety was assessed using the Modified Dental Anxiety Scale (MDAS) [22].

Each session was recorded from the moment when the children and parents entered the office to the end of the treatment. Four short sequences of 2 to 3 min were selected at four key moments: 1—entering the office, 2—the process of being seated and settled in the dental chair, 3—anesthesia (*n* = 11), and 4—treatment.

#### 2.1.3. Analysis

A reliability (test–retest) analysis of all the video sequences was performed by two experts (T.V. and F.D.) using two scales: the modified Venham scale (VS), a hetero-rating scale for young patient behavior, and the newly developed parental hetero-rating scale, used to assess parental behavior.

Following the test–retest analysis of all the cases by both experts, we gathered an observer panel of 20 dental practitioners from different fields, including nine general dentists, five pediatric dentists, three final-year graduate students, two orthodontists, and one implantologist. They were shown three randomly selected video sequences for training purposes. Then, 15 randomly selected cases were analyzed by this panel twice, with a 2-week interval. In order to explore the possible rater effect, the test reliability was calculated for the time point with the highest variation, which was the treatment stage. The test–retest reliability was calculated using Cohen’s Kappa agreement, and the Kendall Tau correlation coefficients was calculated for the inter- and intra-raters. The reliability of all combinations of the 20 raters was calculated using the same tests for the second viewing in the treatment stage.

All the statistical calculations were performed using the SPSS software (IBM SPSS Statistics for Windows, version 27, IBM Corp., Armonk, NY, USA).

### 2.2. Ethics Committee

All subjects gave informed consent for inclusion before they participated in the study. The study was conducted in accordance with the Declaration of Helsinki, and the protocol was approved by the Ethics Committee of CHU Saint-Pierre (O.M.007) on 11 January 2021 (CE/201114; O.M. 007). The protocol is registered on the ClinicalTrials.gov site with the ID NCT05026515 (23 August 2021). The data described in this article are available at: https://zenodo.org/record/5813285 (3 January 2022).

## 3. Results

Sixty-seven children were recruited for this study. They were divided into three groups according to their age (4 years and under, 5 to 6 years, and 7 years and over). Six children were excluded due to refusal to participate, and one refused to allow the recording to be used in the study (see Figure 1).

Of the total of 60 patients, the number of boys was 23 (38%) and the number of girls was 37 (62%). The mean age of the patients was 6.20 years, with a standard deviation of 3.32. The minimum age was 1 and the maximum was 14. Comparable age distributions were observed for the boys and girls, although the girls were slightly older (Chi^2^: *p* = 0.71, Mann–Whitney: *p* = 0.57).

Six patients were too young to answer the MDAS questions, leaving 54 valid cases for this part of the study. There was a slight, non-significant trend of higher anxiety among the boys on average (boys: 10, girls: 8, Mann–Whitney: *p* = 0.14). This difference was in accordance with the negative correlation between the MDAS and age (Kendall’s Tau = −0.23, *p* = 0.027) and the slightly lower age of the boys with respect to the girls in the sample. The sample was predominantly composed of patients with low anxiety (*n* = 30), and there was only one patient with extreme anxiety (see Table 3).

A total of 60 patient/caregiver sequences were observed. The test–retest reliability was found to be high by both experts, ranging from Tau=0.966 to 1. The patterns of extrema were similar between the test and retest. A clear bimodality could be observed for both raters F and V in the treatment stage and for both rating sessions using the parental hetero-assessment scale. This bimodality was not observed in the patient population, but the frequency patterns were similar for both raters throughout the scoring session (see Table 2).

The intra-rater reliability was moderate to high, and the Kendall Tau correlation ranged from 0.513 to 0.852, with a clear tendency of enhancement in the treatment stage compared to the other stages. Note that the variability in the treatment stage was notably higher than that of the other stages (see Table 4).

The Kappa measure of concordance, correcting for probability matching between raters a and b, ranged from 0.513 to 0.588 upon entrance, from 0.579 to 0.750 when sitting in the chair, and from 0.786 to 0.852 during treatment. The interpretation of the scale, according to Fleiss (2003), ranges from “Poor” to “Fair to good” [23]. The better scores were obtained in the treatment stage, where the range of scores was broader. The Kappa scores for the parental hetero-assessment scale were consistently better in the chair and treatment phases than those of the modified Venham scale.

The scores attributed by the 20 practitioners are shown in Figure 2. High scores (2 and up) were not frequently attributed. Parental and children’s scores were globally concordant and higher in the seating, anesthetic and intervention phase.

Combining the scoring of the 20 practitioners, i.e., 380 combinations based on 15 randomly selected patients from the original 60, led to a mean kappa of 0.13 with a standard deviation of 0.18 and median of 0.11, as well as a substantial number of negative scores, including 102 of the 380 combinations of raters (26.8%). When interpreting the scores according to Fleiss, 95.5% of the combinations scored as negative or poor, 3.9% as fair to good, and only 0.5% as excellent.

The test–retest rank correlation Kendall Tau for the 20 practitioners ranged from a non-significant 0.059 to a perfect correspondence of 1.000, with a reasonable mean of 0.658 but a relatively high standard deviation of 0.241. Similarly, the test–retest Spearman’s correlation coefficient for the 20 practitioners ranges from a non-significant 0.066 to a perfect 1.000, with a reasonable mean of 0.696 but a relatively high standard deviation of 0.245.

All the Kendall Tau rank correlation coefficients between the MDAS scores, measuring the patient’s anxiety before entering the office, and anxiety measured by viewing the images averaged for the entrance of the patient and his/her installation in the chair, were significantly positive (*p* < 0.05) for both viewers and both scales, ranging between a weak 0.25 and a substantial 0.41 (see Table 5a). The correlation with the modified Venham scale was systematically higher than that with the parental hetero-assessment scale.

One of the Kendall Tau rank correlation coefficients between the parental hetero-assessment scale upon entrance and the modified Venham scale at treatment was not significant (see Table 5b). All the point estimates were weakly positive (>0.20).

## 4. Discussion

The objective of this study was to propose a scale that can be used to evaluate the behavior of parents on the same order as the modified Venham scale, which is often used in behavioral studies in pediatric dentistry. The strengths of our study were that we built a detailed hetero-evaluation scale and tested it in two forms, and we used a large sample of dentists who participated twice in the randomized video review. We can also mention the fact that all the patients were seen by only two different practitioners, and the fact that all the age categories were equally represented. The evaluation of the child’s level of dental anxiety was performed before entering the treatment room.

The weak points were the non-assessment of the parents’ state of stress before entering the treatment room and the fact that the construction, based on the comparison of the parental scale with the modified Venham scale, was adapted to the parents. The need to combine two categories and obtain an asymmetrical scale did not facilitate the statistical analyses. The construction of our scale met several criteria in Boateng’s 2018 proposal, but it would be interesting to perform a point-by-point analysis [24].

One of the obstacles of the study was the high proportion of non-interventional procedures. When a session does not represent a technical act, the elements to be analyzed are less relevant. The choice of sessions to be filmed should focus on those sessions involving only a technical act, such as a restorative treatment or an extraction, with a measurement of behavior at the time of local anesthesia, particularly high in terms of anxiety, both in children and parents. Even when the results are not significant due to a low number of cases involving administration of local anesthesia, data show that expressed as percentage, parents show a higher proportion of signs of stress and tension (see Figure 2). Possibly, a negative expectation of possible complications during treatment and its negative consequences may play a role in that observation. In this study, heart rate was used as outcome variable, which could be an interesting possibility to be used as additional validation in further studies using video observation [25].

On the technical level, this study was carried out after the COVID-19 health crisis. Currently, parents always wear masks in hospital buildings and in medical offices. It was, therefore, difficult for us to analyze all the facial expressions revealing a possible state of anxiety [26].

In our study, a hetero-rating scale was developed to assess parental behavior as a basis for the study of the relationship between parental behavior and child behavior during treatment. This scale is based on the modified VS. When exploring the psychometric aspects of the modified VS scale, we observed results comparable to those in the literature [8]. The strong point is that this was the first study observing parent/child couples during different stages of treatment. In a previous study on child/parent pairs in a dental situation, Busato et al. [8] collected data based on a questionnaire prior to but not during treatment.

The VS includes a long description of each category. A given description can be confusing and thus can prevent effective and reproducible measurements. While the inter-rater reliability in our study was outstanding, the intra-rater reliability ranged from poor to fair to good, again depending on the stage. On the contrary, despite the recording of three training videos, the 20 observers did not attain the reproducibility achieved by the experts concerning the parental hetero-assessment scale, indicating that greater training and standardization might be necessary. Since the parameters intended to be analyzed using this type of scale are numerous, one could question the applicability of this scale outside a research framework, in which there is a dedicated timeframe and all the attention and training of the evaluator can be exploited, compared to a case when the practitioner is in full action, and the multitude of aspects that must be taken into account to identify a score can cause difficulties. For the dental practice, one might consider scales consisting of triggers to attract the practitioner’s attention based on a limited number of separate aspects or signs to be considered. For research, there are two approaches that can be used to increase the reliability: either the in-depth calibration of each observer or the use of scales consisting of items related to clearly defined aspects and scores (Likert scales). If a rating scale requires extensive calibration, its clinical applicability will be questionable.

According to Ratson et al., children’s behavior is greatly influenced by the attitude of the parents [12]. Parents inform the child’s fears about dental care [11]. Children follow their parents’ reactions in order to better manage a new experience. They are able to perceive whether their parents are calm or anxious with precision and speed by integrating the sensory and emotional information that the latter exhibit without even being aware of it [27]. However, the relationship was not systematically observed in our data, as parental and child behavior were not completely correlated (see Table 4).

However, age is an important exclusion criterion. Indeed, some patients were too young to have sufficient capacity to respond to the MDAS. The first group of children (< 4 years) should have only encompassed children aged 3 and 4. It is from the age of 3 that children develop the ability to understand and solve problems, memorize information, and exercise their own judgment [28].

Parents who understand this impact may decide to use their body and verbal language wisely so as to facilitate the care of their children. When the practitioner senses the inability of the parent to relax their child, certain facilitating options may be offered, such as the decision to behave as a passive observer, thus reducing the pressure on the child. A fair balance must be found between empowering the child and parental support. If support is lacking and the parent withdraws too much, this can have counterproductive effects [10].

The choice to involve a parent during treatment must be a decision made following the analysis of the situation and left to the discretion of the practitioner [29]. Children’s adaptability during dental care is linked to multiple internal factors, such as their age and cognitive and emotional development, and external factors, such as the family environment, socio-economic status and dental experience [4,30].

A negative dental experience can be concrete and/or abstract. It can result from peer-to-peer communication with exaggerated narratives as well as frequent exposure to invasive medical care [8]. A link between the age and the intensity of stress could not be established, as the study was not powered to do so. This aspect should be considered in the future, and the circumstances and types of treatment should be included. Stress stimuli are often concentrated in technical dental procedures, which generally become more invasive with age. On average, it is around the age of 5 years that the first dental treatment takes place and the child is confronted with rotary instruments and a local anesthesia syringe [2].

The number of participants and observers may appear to be a limit, but through randomization, the videos were chosen from a reduced number so that the observers could consciously and diligently judge the entire sample without losing focus.

This study provides an instrument that can be used to study the importance of parental influence on children’s behavior in future research, as well as evidence for its impact. Appropriate knowledge of the dental background of parents’ anxiety could enable better management by the pediatric dentist. It would be interesting to establish the use of questionnaires prior to the first consultation so that the children and parents provide information about their behavior during dental experiences. This could aid the practitioner in the prediction of the attitudes of patients towards treatment [5].

This study could also be extended to other factors, such as the influence of siblings on dental or medical interventions, studying the images and the feelings they project onto each other when they are treated in the same session. Additionally, studies on gendered parental influence are still scarce in the literature [8]. Is anxiety more transmissible through the mother or the father? Furthermore, timing of signs of stress of the parents, child, and dental practitioner can be used to evaluate which of the three parties involved shows first signs of stress in a given situation. However, this would require a more complicated set-up as a precise timing instead of situations would have to be measured.

The implementation of an adapted strategy in cooperation with the parents makes it possible to avoid any conflict that may result from any feeling of exclusion. This prior adjustment can lead to compliance with treatment on the part of the child. It is important to educate parents on behavior management techniques and in particular the choice of positive words both during the session and after. Children need encouragement to maintain their cooperation and even strengthen it. A constructive partnership between the parent and practitioner creates a solid foundation for a positive and long-lasting relationship with the child. In this way, a cycle of fear and negative apprehension can be avoided from early childhood.

## 5. Conclusions

A Venham-type scale with categories involving multiple aspects can be used in research, but the application of such a scale in dental practice requires further development.

The link between parental anxiety and the anxiety of the child was confirmed, but further research is required to incorporate specific aspects of treatment and parent behavior.

## Figures and Tables

**Figure 1 children-10-00249-f001:**
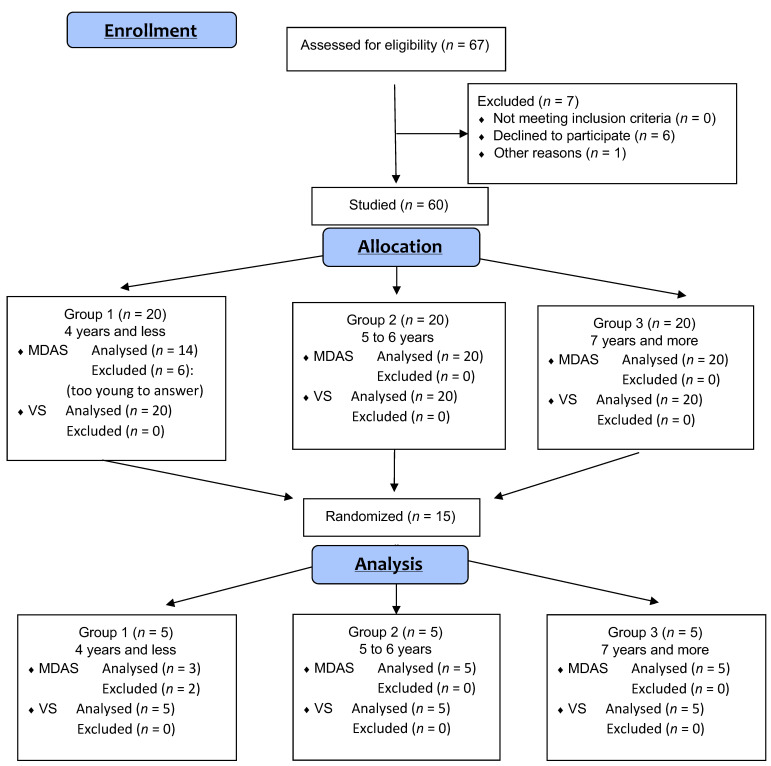
Flow diagram. (MDAS: Modified Dental Anxiety Scale; VS: modified Venham Scale).

**Figure 2 children-10-00249-f002:**
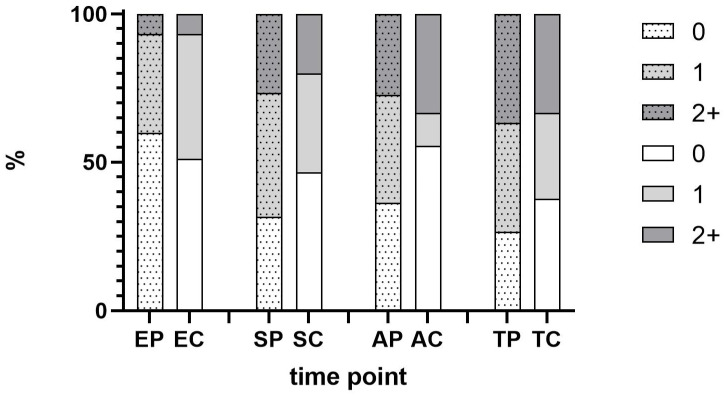
Scores attributed by the 20 practitioners for parents (P: Parents, shaded columns) and children (C: Children, open columns). The scores were given at entrance in the office (E: Entrance), installation and seating (S: Seating), administration of local anesthetic (A: Administration, *n* = 11 cases), and treatment (T: Treatment).

**Table 1 children-10-00249-t001:** Modified Venham scale.

Modified Venham Scale		
Score 0	Relaxed	Smiling, willing, able to converse, display behavior desired by the dentist.
Score 1	Uneasy	Concerned, may protest briefly to indicate discomfort, hands remain down or partially raised. Tense facial expression, high chest. Capable to cooperating.
Score 2	Tense	Tone of voice, question and answers reflect anxiety, during stressful procedure, verbal protest, crying, hands tensed and raised, but not interfering very much. Protest more distracting and troublesome. Child still complies with the request to cooperate.
Score 3	Reluctant	Pronounced verbal protest, crying. Using hands to stop procedure. Treatment proceeds with difficulty.
Score 4	Very disturbed	General crying, body movements sometimes needing physical restraint. Protests disrupt procedure.
Score 5	Totally out of control	Hard loud crying, swearing, screaming. Unable to listen, trying to escape. Physical restraint requires.

**Table 2 children-10-00249-t002:** Parental hetero-assessment scale.

Parental Hetero-Assessment Scale		
Score 0	Relaxed	Open, does not show anxiety, smiling, does not move from the place designated by the practitioner, the tone of voice is calm. Practitioner-patient communication is easy. The dentist redirects patient attention to something pleasant, humoristic, confident (expresses confidence in the dentist), best possible working conditions.
Score 1	Uneasy	Preoccupied, anxious look, impression of fear, worried face, frowning of the eyebrows, attentive and fixed gaze, the tone of voice is slightly distressed.
Score 2	Reluctant	Reluctant to accept the therapeutic situation, worried, agitated, grimaces from time to time, face a little frozen, eyes closed, stands up, tone of voice is distressed (encouragement), repeats what the dentist says.
Score 3	Very disturbed	Frightened look and / or contorted face, gets up and approaches the dental seat, aggressive, moans, sighs, strongly anxious tone (empathy), suggestion to the practitioner, the session is regularly interrupted by protests.
Score 4	Totally out of control	Expression completely frozen, behavior unsuited to the situation, strong agitation, very aggressive (intimidation, criticism), touches the child, leaves the office, opposes the treatment, cries, tears, the practitioner no longer controls the situation, overprotection of the child, impossible to finish the session.

**Table 3 children-10-00249-t003:** Distribution of age and MDAS by gender, and relation of MDAS with age.

	N	Mean	SD	Median	Min	Max	Age 1–4	Age 5–6	Age 7–14			
Age	60	6.20	3.32	5.50	1	14	20	20	20			
Boys	23	5.83	2.90	5.00	2	14	8	9	6	Mann-Whitney U	*z*	*p*
							34.8%	39.1%	26.1%	388.0	0.574	0.6
Girls	37	6.43	3.58	6.00	1	13	12	11	14	Chi^2^	df	*p*
							32.4%	29.7%	37.8%	0.99	2	0.7
MDAS		9.96	4.22	8.50	not anxious	slightly anxious	fairly anxious	very anxious	extremely anxious			
Boys	21	10.95	4.42	10.00	1	11	3	6	0	Mann-Whitney U	*z*	*p*
					4.8%	52.4%	14.3%	28.6%	0.0%	264.5	1.463	0.1
Girls	33	9.33	4.03	8.00	4	19	6	3	1	Chi^2^	df	*p*
					12.1%	57.6%	18.2%	9.1%	3.0%	4.48	4	0.3
Correlation Age-MDAS			95% CI Lower	95% CI Upper	*p*							
	Kendall’s tau_b	−0.23	−0.39	−0.05	0.027							
	Spearman’s rho	−0.31	−0.54	−0.04	0.023							

**Table 4 children-10-00249-t004:** Frequency tables for the parental hetero-assessment scale (a) and the modified Venham scale (b) for the two expert raters and the two scoring sessions (1,2 = Observer 1 or 2; E = Entering the office; C = Taking one’s place in the dental chair; T = Treatment phase). Inter-rater reliability was measured through Cohen’s Kappa and Kendall’s Tau rank-order correlations. The test–retest reliability was measured through Kendall’s Tau. ** Significant at level *a* = 0.001.

(a) Parental Hetero-Assessment Scale							Inter-Rater Reproducibility			
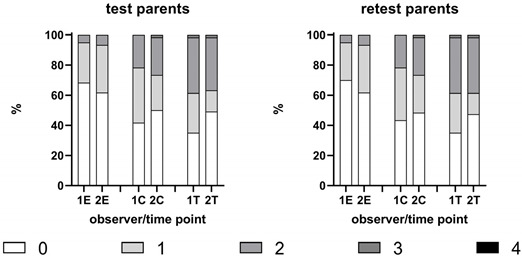							Test			
								Entrance	Chair	Treatment
							Kappa (SE)	0.355 (0.099)	0.617 (0.08)	0.655 (0.079)
							Kendall’s Tau	0.565 **	0.738 **	0.831 **
							Retest			
								Entrance	Chair	Treatment
							Kappa (SE)	0.348 (0.101)	0.589 (0.082)	0.708 (0.075)
							Kendall’s Tau	0.513 **	0.750 **	0.852 **
**Intra-rater**	1E	1C	1T	2E	2C	2T	1E: observer 1, at entering the office 1C: observer1, taking place in dental chair 1T: observer 1, treatment phase Same applies for 2E, 2C and 2T, for observer 2.			
N	60	60	57	60	60	57				
Kendall’s Tau	0.980 **	0.966 **	1.000 **	0.981 **	1.000 **	0.970 **				
**(b) Modified Venham scale**							**Inter-rater reproducibility**			
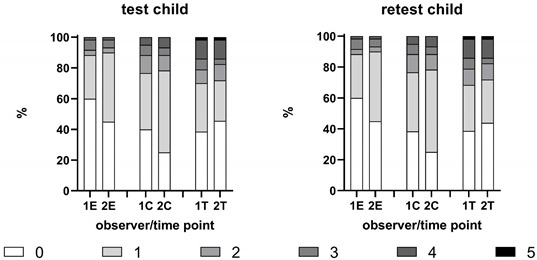							Test			
								Entrance	Chair	Treatment
							Kappa (SE)	0.414 (0.090)	0.368 (0.087)	0.484 (0.083)
							Kendall’s Tau	0.588 **	0.606 **	0.786 **
							Retest			
								Entrance	Chair	Treatment
							Kappa (SE)	0.414 (0.090)	0.340 (0.090)	0.487 (0.082)
							Kendall’s Tau	0.588 **	0.579 **	0.979 **
**Intra-rater**	1E	1C	1T	2E	2C	2T	1E: observer 1, at entering the office 1C: observer1, taking place in dental chair 1T: observer 1, treatment phase Same applies for 2E, 2C and 2T, for observer 2.			
N	60	60	57	60	60	57				
Kendall’s Tau	0.982 **	1.000 **	0.991 **	1.000 **	1.000 **	0.982 **				

**Table 5 children-10-00249-t005:** Rank-order correlation Kendall Tau. (a). Between the raw MDAS scores and the parental hetero-assessment scale (Parent) and the modified Venham scale (Child) averaged for the entrance and installation phases for the two viewers (F and V) and the consecutive views (*n* = 54). (b). Between the parental hetero-assessment scale (Parent) upon entrance and the modified Venham scale (Child) at treatment (*n* = 57) for the two viewers (F and V).

(a) Rank Order Correlation Kendall Tau at Entrance and Installation Phase					
First viewing			95% CI		
	Viewer	Kendall Tau	Lower	Upper	*p*
Child	F	0.412	0.249	0.552	0.000
	V	0.325	0.153	0.478	0.002
Parent	F	0.294	0.120	0.452	0.007
	V	0.258	0.081	0.420	0.017
**Second viewing**					
Child	F	0.404	0.241	0.546	0.000
	V	0.325	0.153	0.478	0.002
Parent	F	0.302	0.128	0.458	0.005
	V	0.283	0.108	0.442	0.009
**(b) Rank order correlation Kendall Tau at entrance and treatment phase**					
**First viewing**			**95% CI**		
	Viewer	Kendall Tau	Lower	Upper	*p*
Child	F	0.268	0.097	0.424	0.025
Parent	V	0.205	0.030	0.368	0.087
**Second viewing**					
Child	F	0.297	0.128	0.450	0.013
Parent	V	0.243	0.070	0.402	0.043

## Data Availability

The protocol is registered on the ClinicalTrials.gov site with the ID NCT05026515. The data described in this article are available at: https://zenodo.org/record/5813285 (3 January 2022).

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
