# Peer review of "A Parental Behavior Scale in Pediatric Dentistry: The Development of an Observational Scale"

_children, 2023, doi:10.3390/children10020249_

Round 1
Reviewer 1 Report
1. Title: the title adequate for the content
2. Abstract is comprehensive by itself
3. References are appropriate and adequate to related works
4. The overall structure of the article is well-organized and relatively well balanced
5. the article is generally written clearly and correctly.
7. the English used in the article is readable but can be improved. sometimes rephrasing a reformulating is necessary.
Evaluating children under 5 was not necessary
Author Response
Dear Reviewer, thank you for all your thoughtful suggestions for corrections. I have applied all of them to my manuscript and you will find them highlighted in yellow.
- Title: the title adequate for the content
- Abstract is comprehensive by itself
- References are appropriate and adequate to related works
- The overall structure of the article is well-organized and relatively well balanced
- the article is generally written clearly and correctly.
- the English used in the article is readable but can be improved. sometimes rephrasing a reformulating is necessary.
Answer: Thank you for your comments. I rephrased a few sentences which, indeed, were not grammatically correct and the manuscript has undergone language edition by a professional service.
Evaluating children under 5 was not necessary
Answer: The evaluation of children under 5 years old interested us because young children are more influenced by the behavior of their parents than older children. However, some children were way too young and couldn't even complete the MDAS. In future study, this element should be taken into consideration.
Best regards
Tania Vanhée
Reviewer 2 Report
This is an interesting study describing the validation process of a new observational scale for caregiver anxiety during pediatric dental treatment. Prior to consideration for publication, the follwing items should be addressed.
1) There are many spelling and grammatical errors throughout the manuscript. There is a need for extensive edits in the English language.
2) As it relates to the study design subsection in the methods section, please clarify and provide a few more details as to who was involved in the expert discussion to develop the scale. Was it the authors? Were other experts invited, and if so, what was their academic/professional background.
3) How was the sample size of 60 determined? Please include your sample size calculation and/or justification. Further, how were the age grouping determined?
4) In the methods section, the term ‘installation on the dental chair’ is a bit confusing. Perhaps the better way to phrase this is ‘the process of being seated and settled in the dental chair’
Author Response
Dear Reviewer, thank you for all your thoughtful suggestions for corrections. I have applied all of them to my manuscript and you will find them highlighted in yellow.
This is an interesting study describing the validation process of a new observational scale for caregiver anxiety during pediatric dental treatment. Prior to consideration for publication, the following items should be addressed.
1) There are many spelling and grammatical errors throughout the manuscript. There is a need for extensive edits in the English language.
Answer: Thank you for your comments. I rephrased a few sentences which, indeed, were not grammatically correct and the manuscript has undergone language edition by a professional service.
2) As it relates to the study design subsection in the methods section, please clarify and provide a few more details as to who was involved in the expert discussion to develop the scale. Was it the authors? Were other experts invited, and if so, what was their academic/professional background.
Answer: The experts who participated in the discussion to develop the scale are among the authors of this article.
3) How was the sample size of 60 determined? Please include your sample size calculation and/or justification. Further, how were the age grouping determined?
Answer: We wanted 3 equal groups of subjects based on age category. For the operating protocol, I referred to a method proposed by a research group in pediatric dentistry at KU Leuven in the field of 2D vs 3D dental radiology (1). You will find this precision and the corresponding reference in the material and method in the manuscript highlighted in yellow.
The choice of this sample is in line with the feasibility of the study. However, we did not issue criteria based on the type of session. We have noticed that routine visits provide less information than sessions involving technical acts, which are more favorable to the observation of various behaviors.
The different age groups are related to the studies we had done previously and to the literature related to the maturity of the child and its link with dental anxiety. (2).
- Van Gorp G, Lambrechts M, Jacobs R, Declerck D. Paediatric dentist’s ability to detect and diagnose dental trauma using 2D versus 3D imaging. Eur Arch Paediatr Dent Off J Eur Acad Paediatr Dent. 2021 Aug;22(4):699–705.
- Vanhee T, Mourali S, Bottenberg P, Jacquet W, Abbeele AV. Stimuli involved in dental anxiety: What are patients afraid of?: A descriptive study. Int J Paediatr Dent. 2020;30(3):276–85.
4) In the methods section, the term ‘installation on the dental chair’ is a bit confusing. Perhaps the better way to phrase this is ‘the process of being seated and settled in the dental chair’
Answer: Thank you for this suggestion, the correction has been made in the manuscript, highlighted in yellow.
Best regards
Tania Vanhée
Round 2
Reviewer 2 Report
Thank you for your attention to the suggested edits. The manuscript is greatly improved.